



# Inadequacies in the representation of sub-seasonal phytoplankton dynamics in Earth system models

Madhavan Girijakumari Keerthi[1,2], Olivier Aumont[1], Lester Kwiatkowski[1], Marina Levy[1]

[1]LOCEAN-IPSL, Sorbonne Université, CNRS, IRD, MNHN, Paris, France

[2]LMD-IPSL, École Normale Supérieure, Université PSL, CNRS, École Polytechnique, Paris, France

*Correspondence to*: Madhavan Girijakumari Keerthi (keerthi.madhavan-girijakumari@locean.ipsl.fr)

**Abstract.** Sub-seasonal phytoplankton dynamics on timescales between 8 days and 3 months significantly contribute to annual fluctuations, making it essential to accurately represent this variability in ocean models to avoid distorting long-term trends. This study assesses the capability of Earth System Models (ESMs) participating in the Coupled Model Intercomparison Project

Phase 6 (CMIP6) to reproduce sub-seasonal surface ocean phytoplankton variations observed in ocean color satellite data. Our findings reveal that, unlike sea surface temperature, all models struggle to accurately reproduce the total surface ocean phytoplankton variance and its decomposition across sub-seasonal, seasonal, and multi-annual timescales. Over the historical period, some models strongly overestimate sub-seasonal variance and exaggerate its role in annual fluctuations, while others underestimate it. Our analysis suggest that underestimation of sub-seasonal variance is likely a consequence of the coarse

horizontal resolution of CMIP6 models, which is insufficient to resolve mesoscale processes—a limitation potentially alleviated with higher-resolution models. Conversely, we suggest that the overestimation of sub-seasonal variance is potentially the consequence of intrinsic oscillations such as predator-prey oscillations in certain biogeochemical models. ESMs consistently show a reduction in variance at sub-seasonal and seasonal timescales during the 21[st] century under high-emission scenarios. The poor capability of CMIP6 models at simulating sub-seasonal chlorophyll dynamics casts doubt on their

projections at these temporal scales and multi-annual timescales. This study underscores the need to enhance spatial resolution and constrain intrinsic biogeochemical oscillations to improve projections of ocean phytoplankton dynamics.

## 1 Introduction

Phytoplankton, the photoautotrophic microscopic organisms populating the upper layers of the ocean, form the base of marine food webs and play a crucial role in driving ocean biogeochemical cycles. Over recent decades, climate change due to

anthropogenic activities has emerged as a significant threat to ocean phytoplankton, altering the key environmental factors essential for their growth and survival (Behrenfeld et al., 2006; Bindoff et al., 2019). The repercussions extend beyond the



marine environment, impacting the global carbon cycle and the future absorption of atmospheric carbon dioxide by the ocean (Bopp et al., 2005; Gregg et al., 2005).

Earth system models (ESMs) are indispensable tools for forecasting the impacts of climate change on ocean primary productivity, and comprehending the intricate interplay between oceanic physical and biological processes. ESMs consistently project increased stratification across various climate change scenarios, enhancing phytoplankton nutrient limitation in low-latitude oceans (Steinacher et al., 2010; Bopp et al., 2013; Krumhardt et al., 2017; Kwiatkowski et al., 2017; Moore et al., 2018). As a consequence, marine primary production is globally projected to decrease (Sarmiento et al., 2004; Cabré et al.,

2014; Fu et al., 2016; Kwiatkowski et al., 2020). However, the extent of this decline remains highly uncertain across model ensembles, including uncertainty in even the direction of change (Bopp et al., 2013; Krumhardt et al., 2017; Kwiatkowski et al., 2020).

A comprehensive comparison of the ocean biogeochemistry simulated by ESMs with observations can shed light on model

deficiencies and associated driving factors (Séférian et al., 2020; Kwiatkowski et al., 2018; Kessler & Tjiputra, 2016; Planchat et al., 2023). The availability of two decades of daily satellite ocean color measurements of surface chlorophyll (SChl, a proxy for phytoplankton biomass) at global scale represents a unique means to evaluate the skill of ESMs to simulate phytoplankton. However, assessing multi-model uncertainty in climate projections has to go beyond evaluating solely the model mean state performance. It is crucial to assess models against observed variations across all timescales to bolster confidence in their

projections (Séférian et al., 2020). This is particularly critical for phytoplankton as it is characterized by large natural variability at diverse timescales, which often masks the long-term trends (Henson et al., 2010; Henson et al., 2016; Doney et al., 2014; Keerthi et al., 2022).

The seasonal cycle represents the primary mode of SChl variability (Demarcq et al., 2012). However, in many oceanic regions,

sub-seasonal variability is equally significant and occasionally surpasses seasonal fluctuations (Keerthi et al., 2022; Prend et al., 2022; Levy et al., 2024). Sub-seasonal variability comprises high-frequency fluctuations associated with sub-seasonal atmospheric variability including storms and tropical cyclones (Carranza et al., 2015), sub-seasonal climate modes (Resplandy et al., 2009), mesoscale and submesoscale eddies (Gaube et al., 2014), and intrinsic biological processes (Mayersohn et al., 2021). In various locations, phytoplankton variations at sub-seasonal frequencies can be more than two times as large as the

climatological mean (Resplandy et al., 2009; Thomalla et al., 2011; Keerthi et al., 2021). In contrast, low-frequency (multi-annual) variations with distinct regional characteristics are evident, and correlated with large-scale climate modes (Wilson and Adamec, 2001; Racault et al., 2017; Park et al., 2018; Resplandy et al., 2009; Lovenduski and Gruber, 2005; Martinez et al., 2016). But with the exception of specific tropical regions, their contribution to total variability remains relatively modest (Keerthi et al., 2022).




Previous studies on simulated ocean primary production have predominantly focused on evaluating the mean state performance of models (Séférian et al., 2020; Bopp et al., 2013; Kwiatkowski et al., 2020), neglecting a comprehensive exploration of different temporal scales in model assessments. Capitalizing on high frequency global measurements of satellite ocean color SChl, we evaluated the performance of historical simulations produced by ESMs participating in the Coupled Model
Intercomparison Project Phase 6 (CMIP6) to simulate global surface ocean phytoplankton dynamics across diverse temporal scales (sub-seasonal, seasonal, and multi-annual), with a specific focus on high frequency sub-seasonal variability. To do so, we applied the temporal decomposition methodology developed for SChl satellite data in Keerthi et al. (2022) to CMIP6 historical simulations. Our analysis of SChl is additionally contrasted with that of sea surface temperature (SST), a typically well-simulated physical ocean parameter, particularly in comparison to SChl.


Satellite ocean color measurements of SChl reveal that the cumulative effect of high-frequency sub-seasonal fluctuations can modulate year-to-year variations of SChl, a factor that has historically been overlooked (Keerthi et al., 2022; Prend et al., 2022). The changing frequency of extreme atmospheric events, such as marine heatwaves (Frolicher et al., 2018) and tropical cyclones (Knutson et al., 2020; Walsh et al., 2016), coupled with mesoscale and submesoscale variability linked to global
warming scenarios (Martínez-Moreno et al., 2021), may actively contribute to alter the sub-seasonal variability of SChl. The intricate interplay between the different timescales has therefore the potential to shape overarching long-term trends in surface ocean phytoplankton and thus deserves a specific focus. We therefore extend our analysis to future model projections using simulations of the high-emission scenario SSP5-8.5.

The enhancement of resolution in coupled climate models improves atmospheric and oceanic dynamics, thereby reducing biases in the mean state and variability of various quantities (Muller et al., 2018). In our analysis of CMIP6 simulations, we also used the opportunity to compare the performance of a higher-resolution model version (MPI-ESM1.2-HR) and its lower-resolution counterpart (MPI-ESM1.2-LR) in simulating SChl variability across different timescales. MPI-ESM1.2-HR has a horizontal resolution twice as high for the atmospheric component (100 km) and more than twice as high for the oceanic
component (~40 km) compared to MPI-ESM1.2-LR (200 km and 150 km for the atmospheric and oceanic components, respectively).

## 2 Data and Methods

**Observation data:** We utilised the datasets outlined in Keerthi et al., (2022) for observed SChl and SST. The SChl data is the Level 3 Mapped 9x9 km resolution 8-day averaged product (release 4.1), covering the period from January 1998 to December
2014. This dataset was obtained from the European Space Agency Ocean Color Climate Change Initiative (ESA OC-CCI; Sathyendranath and Krasemann, 2014) and can be accessed at http://www.oceancolour.org/. The product is a merged compilation from various ocean color satellite missions, including the Moderate Resolution Imaging Spectroradiometer



(MODIS)-Aqua, the Sea-Viewing Wide Field-of-View Sensor (SeaWiFS) and the Medium Resolution Imaging Spectrometer (MERIS). Given the limited coverage of the satellite-derived SChl data in polar regions, our analysis is concentrated on the region between 60°S and 60°N.

For SST, we used the daily 25x25 km resolution Optimum Interpolation Sea Surface Temperature (OISST) data, spanning from January 1998 to December 2014. This dataset is accessible through the National Oceanic and Atmospheric Administration (NOAA) at https://www.ncdc.noaa.gov/oisst/optimum-interpolation-sea-surface-temperature-oisst-v20). The OISST data integrates observations from satellites, ships, buoys, and Argo floats.

| CMIP6 Simulations | Physical Ocean Model | Ocean BGC Model | Horizontal resolution (Physical & BGC Model) | Model Simulations | References |
|---|---|---|---|---|---|
| IPSL-CM6A-LR | NEMO-OPA | PISCES | 100 km | Historical | Boucher et al. 2018, 2021 ; Séférian, 2018 |
| IPSL-CM6A-LR-INCA | | | | | |
| CNRM-ESM2-1 | | | | | |
| CESM2 | POP2 | MARBL | 100 km | Historical | Danabasoglu, 2019a, b, c; |
| CESM2-FV2 | | | | | |
| CESM2-WACCM-FV2 | | | | | |
| MPI-ESM1.2-HAM | MPIOM | HAMOCC6 | 150 km | Historical | Neubauer et al., 2019 ; Wieners,et al., 2019a, b, c ; Jungclaus et al., 2019a, b ; Schupfner et al., 2019 |
| MPI-ESM1.2-LR | | | | Historical, SSP5-8.5, piControl | |
| MPI-ESM1.2-HR | MPIOM | HAMOCC6 | 40 km | | |
| NorESM2-LM | MICOM | HAMOCC | 100 km | Historical, SSP5-8.5 piControl | Seland et al., 2019a, b, c ; Bentsen et al., 2019a, b ; |
| NorESM2-MM | | | | | |

**Table 1.** The CMIP6 Earth system models used in this study; their individual components used to represent ocean and marine biogeochemistry; nominal horizontal resolutions of their ocean and marine biogeochemical models; simulations that were assessed.





**CMIP6 Historical Simulations**: We obtained SChl and SST data for the period 1981-2014 from https://esgf-
node.ipsl.upmc.fr/search/cmip6-ipsl/. Our analysis focused on the 11 CMIP6 historical simulations that had daily SChl outputs.
The horizontal nominal resolution for ocean dynamics in most models is 100 km, except for MPI-ESM1-2-LR, MPI-ESM1.2-
HAM and MPI-ESM1-2-HR, which have a resolution of 150, 150 and 40 km, respectively. For SST analysis, NorESM2-LM
and NorESM2-MM are excluded due to the absence of daily outputs. The ensemble member "r1i1p1f1" is utilized for all
models, except for CNRM-ESM2-1, where "r1i1p1f2" is used. Details relating to the eleven models utilized in our study are
provided in Table 1.

**CMIP6 Future projections**: To project future variability in SChl at various timescales, we utilized a subset of ESMs (MPI-
ESM1.2-LR, MPI-ESM1.2-HR, NorESM2-LM and NorESM2-MM) that performed the SSP5-8.5 scenario, providing daily
resolution data for the period 2084-2100. Pre-industrial control simulations (piControl) were used to ensure that observed
climate change signals were not influenced by model drift. Analysis of piControl simulations is presented for MPI-ESM1.2-
LR, MPI-ESM1.2-HR and NorESM2-LM only, as NorESM2-MM does not provide daily SChl outputs in piControl
simulations.

All analyses were performed on satellite observations and CMIP6 simulations regridded on a common 1°x1° spatial grid and
a temporal resolution of 8 days. Satellite observations were regridded using area-weighted averaging. CMIP6 simulations were
transformed using the CDO remapping tool remapdis.

**Temporal decomposition and variance explained:** We applied a decomposition methodology akin to that in Keerthi et al.,
(2020, 2022) and Vantrepotte & Mélin (2009, 2011), to decompose the SChl and SST timeseries at each grid point to seasonal
($S_t$), multi-annual ($MA_t$) and sub-seasonal ($SS_t$) components. A comprehensive description of this methodology is available in
Keerthi et al., (2020, 2022). This decomposition ensures that at every geographical location, the total time series ($X_t$) can be
expressed as the sum of its sub-components: $X_t = SS_t + S_t + MA_t$. The seasonal component ($S_t$) encapsulates variability within a
period of 3 months to 1 year as well as year-to-year variations in the seasonal cycle. The multi-annual component ($MA_t$)
represents variability with a timescale longer than 8 months, while the sub-seasonal component ($SS_t$) captures variability with
a period shorter than 88 days along with any irregular variability outside of that specified range. This method allows for small
overlaps in the frequency ranges associated with each component.

The total variance of the SChl and SST timeseries can be decomposed into the cumulative variance explained by its different
components along with the covariance amongst these components. In practice, the covariance terms are generally negligible.
The proportional contribution of each component to the total variance is expressed as a percentage.





**Spatial scale of coherence:** The spatial scale of coherence associated with each time component (seasonal, multi-annual, and sub-seasonal) is defined as the extent over which the temporal signal remains self-coherent. We conducted cross-correlation analyses by comparing the decomposed time series of all grid cells included in a disk with a diameter of 2400 km. This sets

then an upper limit of 2400 km to the scale we can infer with this method. We then counted the number of grid cells where the cross-correlation exceeded 0.8 and converted this count into a distance measurement. The chosen threshold value of 0.8 aligns with that in Keerthi et al., (2020, 2022).

**Spatial decomposition**: To assess the relative contribution of spatial scales at intervals of 100 km, 200 km, and so forth, to

the sub-seasonal signal, we executed a spatial decomposition at every 8-day time step. This decomposition methodology is based on an iterative application of the heat diffusion equation, as presented in Weaver and Courtier (1990), that has been previously implemented in the work of Keerthi et al., (2013, 2016).

## 3 Results and Discussions

### 3.1 Evaluation of the mean state

Before turning to the analysis of temporal variability simulated by the models, we initially compare the mean state of the models with satellite-derived SChl. It is important to note that, due to our specific model selection process, the ensemble mean state presented here may differ from standard CMIP6 analyses, which typically include a wider selection of models. Our study focuses solely on simulations providing daily SChl outputs, discarding models that do not meet this criterion.

The ensemble mean of SChl from 11 CMIP6 simulations, as detailed in Table 1, is compared with satellite-based estimates derived from ESA OC-CCI Ocean color data (Figure 1a). Key features include elevated SChl levels in temperate, subpolar, and upwelling regions, contrasting with notably lower levels in the subtropical gyres. The latter areas are characterized by consistently low-nutrient conditions, while the former receive intermittent nutrient influxes through upwelling or deep mixing. Although the CMIP6 ensemble mean generally aligns with observations, there is a notable overestimation across the entire

ocean (Figure 1a, b).

Séférian et al., (2020) undertook a comparison between the mean state of CMIP6 simulations and satellite SChl measurements (ESA OC-CCI) also spanning 1998-2014. Their results indicate significant discrepancies between models and observational data in reproducing the SChl mean state. The models assessed in both studies are CESM2, CNRM-ESM2-1, IPSL-CM6A-LR,

MPI-ESM1.2-LR, and NorESM2-LR. Their findings suggest MPI-ESM1.2-LR persistently and globally overestimates SChl. NorESM2-LR slightly overestimates SChl in the tropics and subtropics but underestimates it in polar regions. CESM2, CNRM-ESM2-1, and IPSL-CM6A-LR displays varying biases relative to satellite SChl across regions.



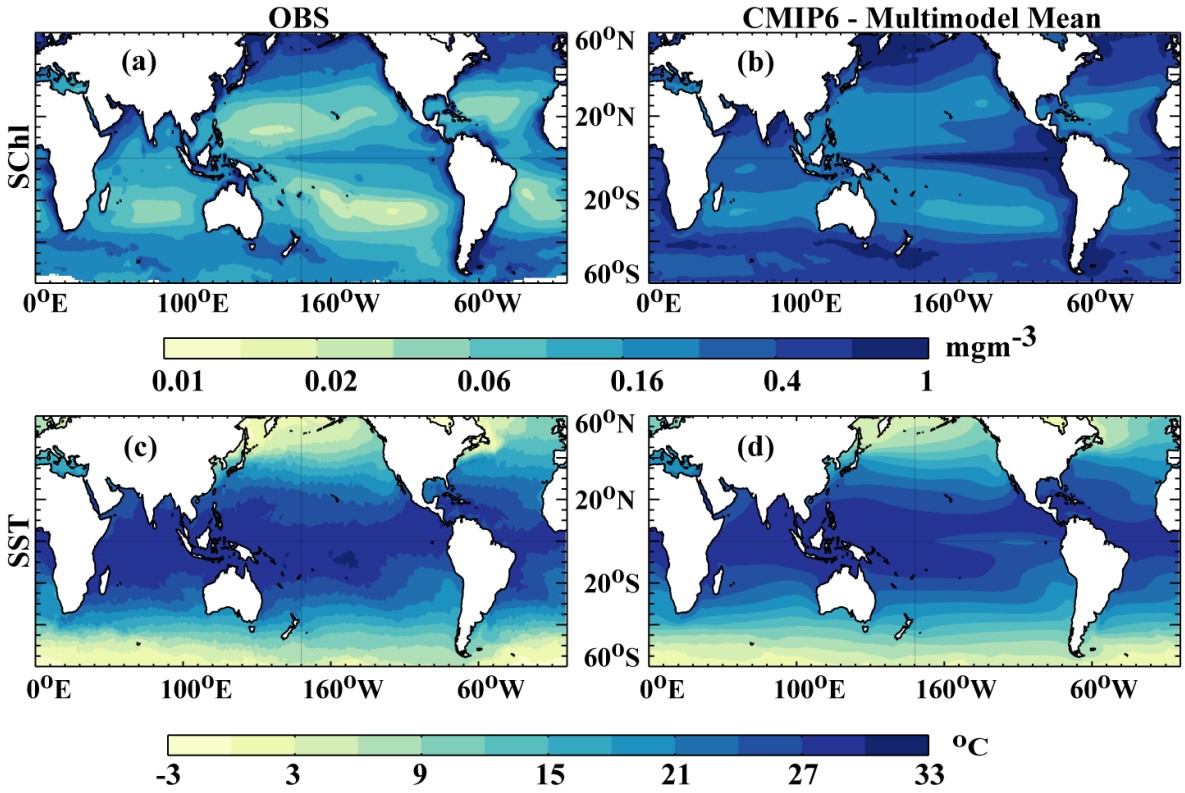

**Figure 1: Mean state evaluation:** Annual mean SChl (a) Observed (ESA OC-CCI product) and (b) CMIP6 multi-model mean for the years 1998-2014 and domain 60°N-60°S. (c & d) Similarly for SST.

The ability of the different model configurations to represent spatial variability is quantified in the Taylor diagram (Fig. 2a). This analysis reveals important differences between models. Most models analyzed here systematically underestimate the observed SChl spatial variance. All models, with the exception of the MPI models, exhibit weak spatial variability ranging from 0.15 to 0.3 mg Chl/m³, compared to 0.60 mg Chl/m³ in satellite observations. MPI models show a similar spatial variability to the satellite observations, ranging from 0.6 to 0.7 mg Chl/m³. The spatial correlation between CMIP6 models and observations remains below 0.6 (Fig. 2a), with MPI models showing particularly low correlations, below 0.2.

In agreement with Séférian et al., (2020), models sharing a common physical ocean model generally have similar skill, though exceptions are noted for CNRM-ESM2-1 and MPI-ESM1.2-HR. CNRM-ESM2-1, which, like IPSL-CM6A-LR and IPSL-CM6A-LR-INCA, includes the coupled physical biogeochemical model NEMO-PISCES, shows a slightly higher spatial standard deviation than the IPSL models. MPI-ESM1.2-HR, which shares the same physical and biogeochemical model MPIOM-HAMOCC as MPI-ESM1.2-HAM and MPI-ESM1.2-LR, exhibits a higher spatial standard deviation. CESM and





NorESM2 configurations, which respectively use the coupled physical biogeochemical models POP2-MARBL and MICOM-HAMOCC, simulate similar spatial correlations. Despite MPI and NorESM2 models using the same ocean biogeochemical model HAMOCC, there are notable differences in their simulated spatial standard deviations. However, the spatial correlation with observations remains consistent among these models.

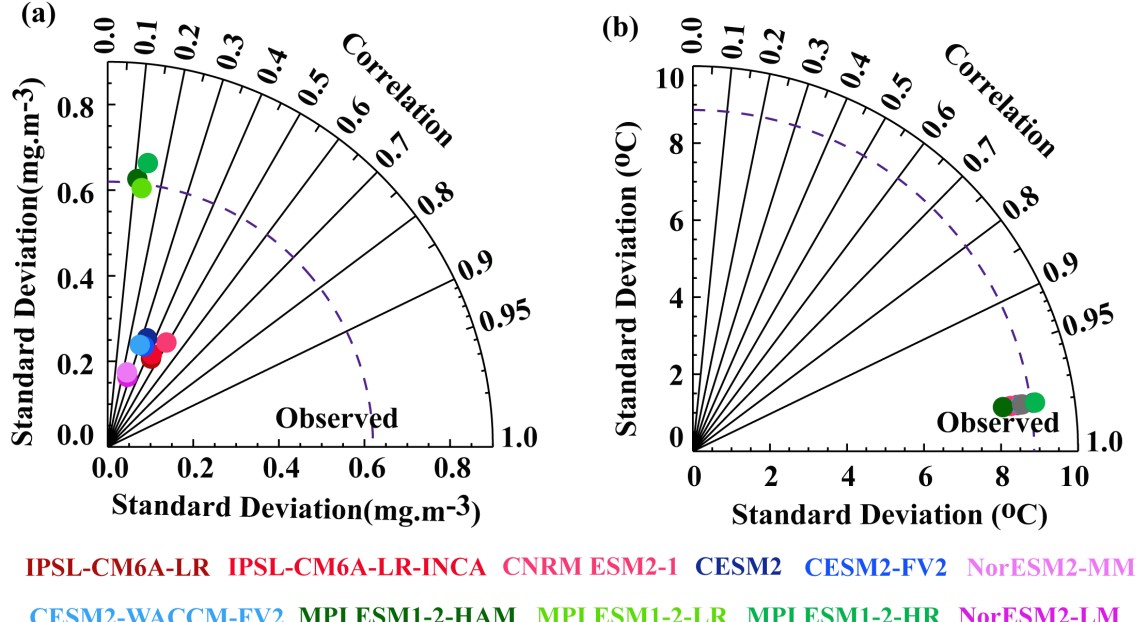

**Figure 2: Evaluation of the mean spatial distribution:** Taylor diagram for the annual mean (a) SChl and (b) SST over the years 1998-2014 and domain 60N-60S.

A comparison of model ensemble mean SST with observed SST reveals that model ensemble mean and observations exhibit
a similar spatial variability, both in terms of amplitude and patterns (Figure 1c, d). All models display a particularly high correlation (>0.95) as well as comparable standard deviations (8-9°C) to the satellite observations (9°C) (Figure 2b). Among the models, MPI-ESM1.2-HAM is positioned at the outer boundary with a spatial standard deviation of approximately 8°C. In conclusion, all CMIP6 models examined here achieve a much better agreement with the observed spatial patterns of SST than SChl.

**3.2 Exploring differences in model performance across temporal scales**

Here, we assess the capability of each CMIP6 model to capture the variability of SChl across different timescales (seasonal, sub-seasonal, and multi-annual), as defined in Section 2. Figures 3a and 3b illustrate the SChl variance across these timescales and their respective contributions to the overall SChl variance, averaged globally, in comparison with satellite-derived SChl



observations. Our analysis reveals that, for satellite-derived SChl, seasonal variability demonstrates the highest normalised standard deviation (~0.3), followed by sub-seasonal variability (~0.2), and then multi-annual variability (~0.15). The relative contribution of these timescales to the overall SChl variance mirrors this pattern, with seasonal variability accounting for approximately half of the total variance, while sub-seasonal and multi-annual variability contribute 30% and 20% respectively. Despite being based on the same satellite SChl dataset as in Keerthi et al., (2022), we observe a notable reduction in the relative contribution of sub-seasonal variability to the total SChl variance in the current satellite SChl product. This difference is discussed in section 3.4.

The variability of SChl across different timescales varies significantly among the CMIP6 simulations. With the exception of IPSL-CM6A-LR, IPSL-CM6A-LR-INCA, and CNRM-ESM2-1, most models overestimate the observed standard deviation at both seasonal and sub-seasonal timescales, often exceeding it by a factor of 3. With the exception of MPI-ESM1.2-HAM and MPI-ESM1.2-LR, the distribution of standard deviation among the three defined timescales resembles that observed in satellite SChl: variability at seasonal timescales is the largest followed by sub-seasonal variability and then multi-annual. In contrast, MPI-ESM1.2-HAM and MPI-ESM1.2-LR exhibit the largest variance at sub-seasonal timescales. In the case of IPSL-CM6A-LR and IPSL-CM6A-LR-INCA, there is a slight overestimation of the standard deviation at seasonal timescales and conversely a slight underestimation at sub-seasonal timescales. Multi-annual variance, compared to other timescales, is relatively similar between models. CESM2, CESM2-FV2, CESM2-WACCM-FV2, MPI-ESM1.2-HAM, and MPI-ESM1.2-LR slightly overestimate the observed variance at the multi-annual timescale, whereas CNRM-ESM2-1, NorESM2-LM, and NorESM2-MM models show a slight underestimation.

To compare the relative importance of each component, we calculated the normalized standard deviation for each time series component ($SS_t$, $S_t$, $MA_t$). This normalization allows for a standardized comparison across different locations and variables, providing insight into the dominant modes of variability in the SChl and SST time series.

When examining the relative contribution of each component to the total variance, differences between models are more apparent. With the potential exception of MPI-ESM1.2-HR, none of the models accurately replicate the observed decomposition. IPSL-CM6A-LR, IPSL-CM6A-LR-INCA, and CNRM-ESM2-1 overestimate the variance attributed to the seasonal timescale (60-70%), while the remaining 30-40% is evenly distributed between the sub-seasonal and multi-annual components. Conversely, CESM2, CESM2-FV2, CESM2-WACCM-FV2, MPI-ESM1.2-HAM, MPI-ESM1.2-LR, NorESM2-LM, and NorESM2-MM overestimate the relative contribution of sub-seasonal variability (40-50%) and consistently underestimate the contribution of the multi-annual timescale (5-15%). In these simulations, both seasonal and sub-seasonal variations contribute approximately equally to the total variance, deviating from the observed patterns.



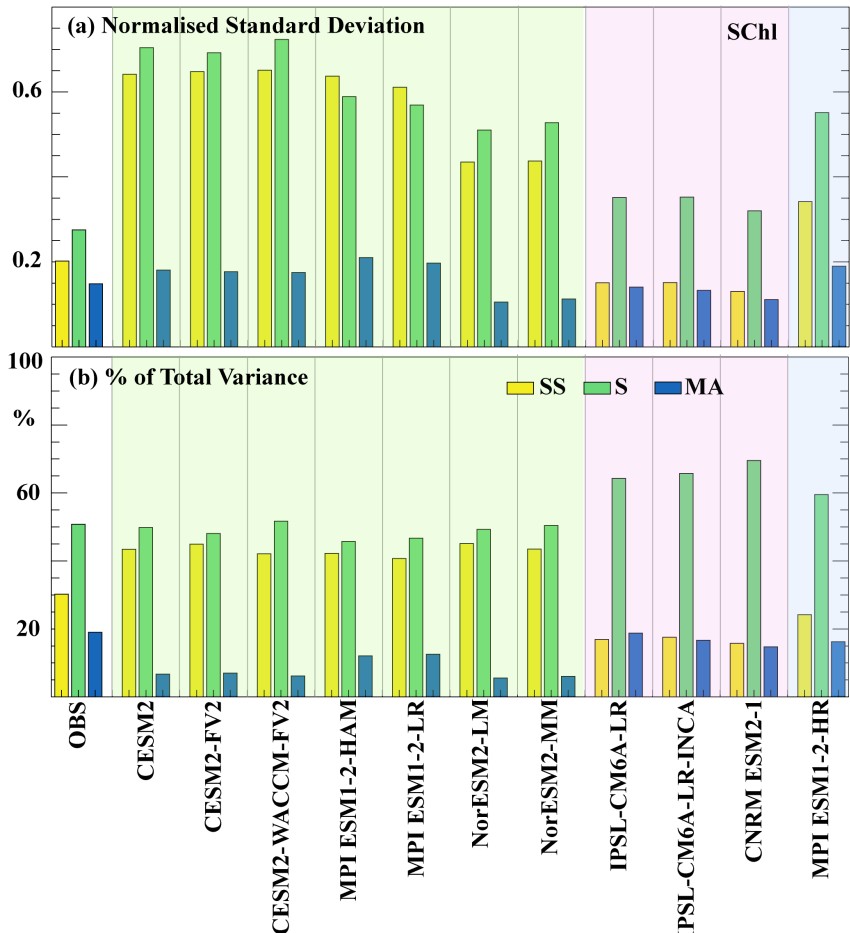

**Figure 3: Variability across timescales for SChl** : (Left Panel) (a) Normalised standard deviation of SChl from observation and CMIP6 historical models. Standard deviation at each grid point is normalised by the mean over each grid. (b) Percentage of SChl variance explained by each component (sub-seasonal, seasonal and multi-annual) for observations and CMIP6 historical models. Shading represents the different model groups described in Section 3.2, with green for Group 1, pink for Group 2, and blue for Group 3.

The CMIP6 SChl simulations can be broadly categorized into three distinct groups based on their performance in capturing SChl temporal variability:

**1. Overestimation of sub-seasonal variability**: Models falling into this category, including CESM2, CESM2-FV2, CESM2-WACCM-FV2, MPI-ESM1.2-LR, MPI-ESM1.2-HAM, NorESM-LM, and NorESM-MM, predominantly overestimate the relative contribution of sub-seasonal variance to the total variance. Consequently, the relative contribution of seasonal and



multi-annual timescales is strongly underestimated. They strongly overestimate the observed SChl standard deviation by approximately threefold at both seasonal and sub-seasonal timescales.

**2. Underestimation of sub-seasonal variability**: The models in this category, which are IPSL-CM6A-LR, IPSL-CM6A-LR-INCA, and CNRM-ESM2-1, underestimate both the variance at sub-seasonal timescales and its relative contribution to the

total variance. Nevertheless, they approximately reproduce the observed total variance because the variance at seasonal timescales and its relative contribution to total variance are both overestimated.

**3. Overestimation of total variance but consistent temporal decomposition:** The only model in this category is MPI-ESM1.2-HR. This model correctly simulates the relative contribution of the three considered timescales to the total variance.

However, the variances are strongly overestimated.

The standard deviation across different timescales and the relative contribution of these timescales to the total SST variance show distinct patterns compared to SChl (Figure 4). As for SChl, the primary driver of natural variability in SST is the seasonal cycle (Keerthi et al., 2022). In observations, the seasonal cycle exhibits the largest standard deviation (0.19) and accounts for

approximately 80% of the total SST variance. Multi-annual variability has a standard deviation of 0.05 and explains around 10-12% of the total variance, while sub-seasonal variability is characterized by the lowest standard deviation (0.04) and contributes the least to the total variance (<10%). The multi-annual component makes a relatively minor contribution everywhere, except in the tropics where it is largely related to ENSO (Keerthi et al., 2022). Sub-seasonal variability has a minimal impact on the total SST variance everywhere in the ocean (Keerthi et al., 2022).


Consistent with observations, all simulations exhibit the highest SST standard deviation at the seasonal timescale, followed by the multi-annual and then the sub-seasonal timescale. Across all simulations, seasonal variability accounts for approximately 80% of the total variance, followed by the multi-annual component (~10%). In contrast to SChl, the standard deviation of the sub-seasonal and multi-annual timescales and their relative contribution to the total variance show minimal differences

between models and closely resemble observations. All simulations slightly overestimate the standard deviation at the seasonal timescale and its relative contribution (by approximately 5%) whereas both are slightly underestimated at sub-seasonal timescales (by about 5%). Among the models, the IPSL-CM6A-LR and MPI-ESM1.2-LR show a larger overestimation of the observed standard deviation at both seasonal and multi-annual timescales.





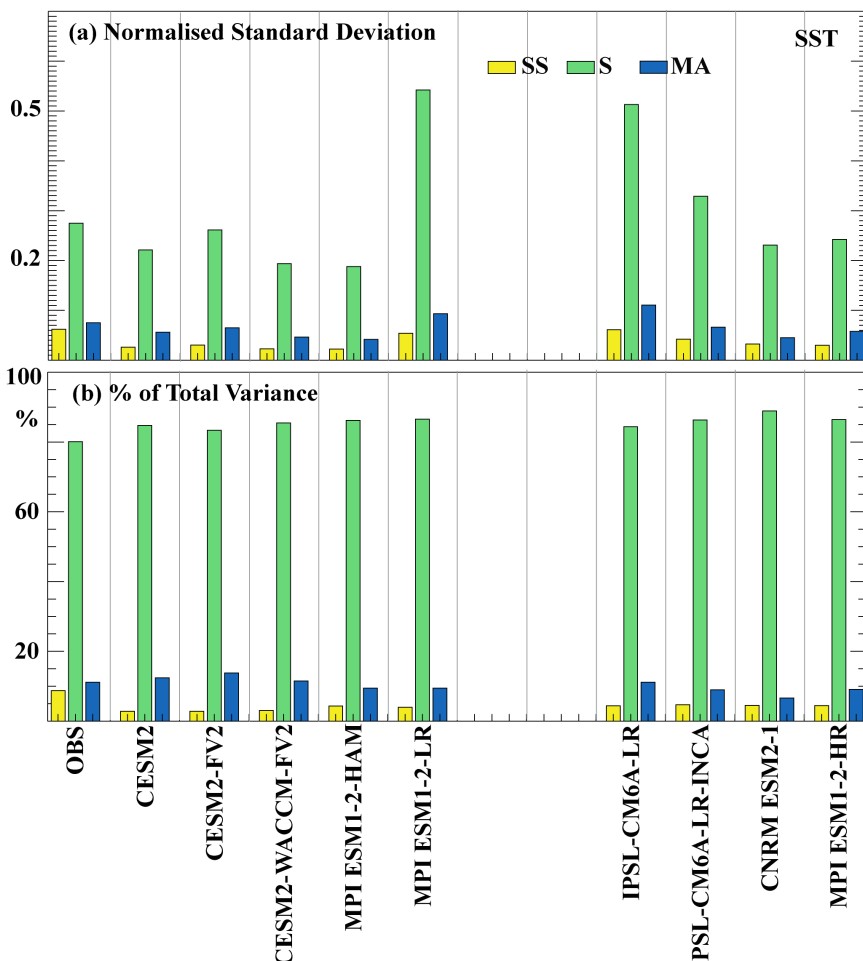

**Figure 4: Variability across timescales for SST**. Similar to Figure 3, but for SST.

## 3.3 Spatial scales corresponding to timescales of variability

The evaluation of spatial scales provides insights into the distance over which a signal remains coherent. This helps identify the relevant driving processes at each timescale, aiding understanding of the differences between models and observations.

In the satellite-derived observations of SChl, the seasonal component displays the largest spatial scales, between 500 and 1500 km (Figure 5a, b). These scales are coherent with factors driving seasonality, such as variations in surface stratification and solar irradiance which operate at basin scales. Likewise, in all CMIP6 simulations, the largest spatial scales (>~800 km) correspond to the seasonal cycle. Two groups can be broadly identified among the models (Figure 5a,b). The first group, that includes CESM2, CESM2-FV2, CESM2-WACCM-FV2, NorESM2-LM, and NorESM2-MM, is characterized by the largest spatial scales, approximately ~1500 km. In contrast, in the second group that comprises IPSL-CM6A-LR, IPSL-CM6A-LR-



INCA, CNRM-ESM2-1, MPI-ESM1.2-HAM, MPI-ESM1.2-LR and MPI-ESM1.2-HR, spatial scales associated to seasonal variability are smaller, ~1000 km. Similarly, the seasonal cycle of SST is characterized by very large spatial scales exceeding 2000 km which are even greater than those of SChl (Figure 5c, d). Indeed, both in the models and the observations, the

computed scales reach the upper limit of 2400 km we set in our methodology which is not the case for SChl.

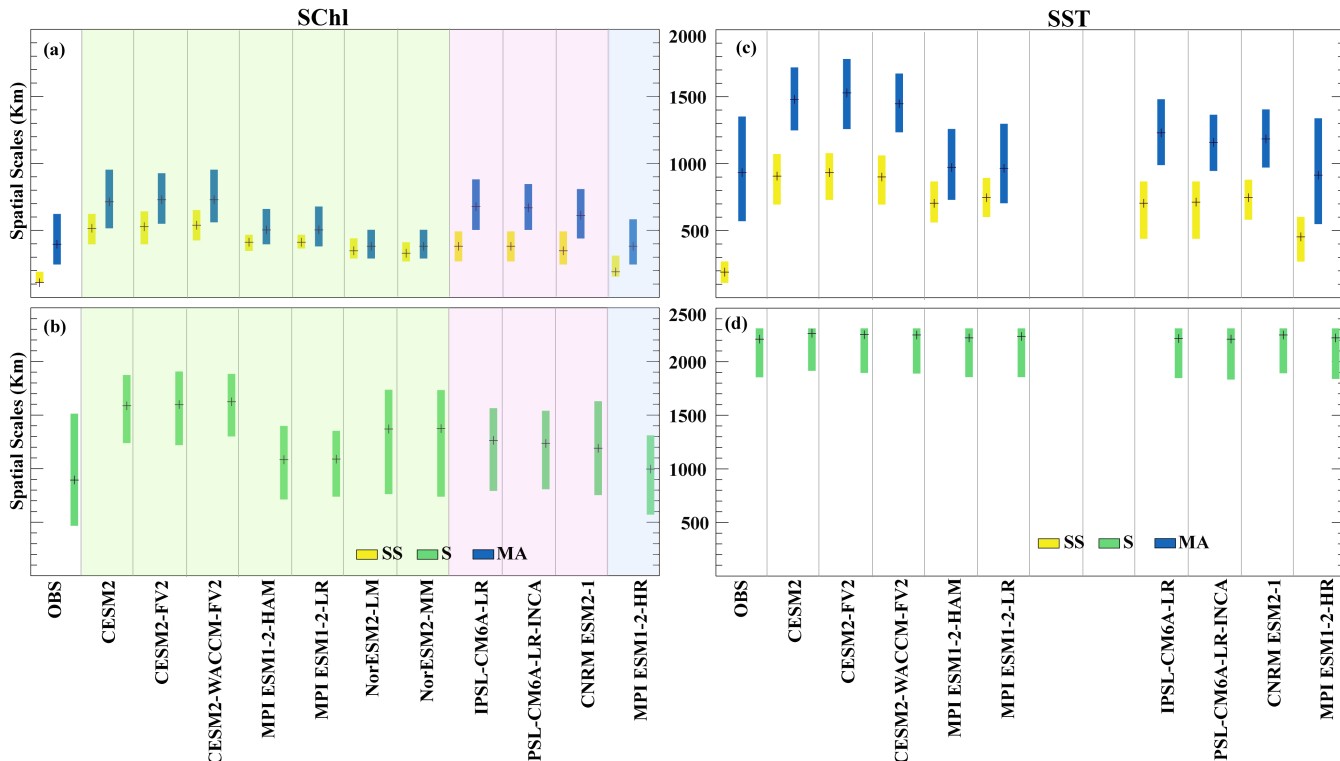

**Figure 5: Spatial scales corresponding to each timescales: The** spatial scales associated with sub-seasonal (yellow), seasonal (green) and multi-annual (blue) variations of SChl (a,b) and SST (c,d). The black line within each box denotes the

median. Shading in panel a and b represents the different model groups described in Section 3.2, with green for Group 1, pink for Group 2, and blue for Group 3.

For both SChl and SST, the spatial scales corresponding to multi-annual variability are the second largest (Figure 5 a,c). Satellite SChl measurements associate multi-annual variability with spatial scales ranging from about 300 to about 600 km

with a median close to 400 km, aligning with climate mode scales on average. However, in the tropics, where this temporal component dominates, spatial scales can extend beyond these averages (Keerthi et al., 2022). In CMIP6 simulations, the spatial scales simulated for this component vary among models. IPSL-CM6A-LR, IPSL-CM6A-LR-INCA, CNRM-ESM2-1, CESM2, CESM2-FV2, and CESM2-WACCM-FV2 simulate the largest values with a median close to about 700 km, whereas MPI-ESM1.2-HAM, MPI-ESM1.2-LR, MPI-ESM1.2-HR, NorESM2-LM, and NorESM2-MM have comparatively smaller



spatial scales, with a median close to that of the observations (~400-500 km). For SST, observational data show spatial scales of about 900 km, while CMIP6 simulations display a broader range from ~500-1800 km. This variability in models may stem from averaging across regions where multi-annual variability is not dominant, given that for both SST and SChl, multi-annual variability is primarily significant only over the equatorial Pacific and Indian Ocean where ENSO is dominant (Keerthi et al., 2022). Additionally, models generally display broader and more intense ENSO patterns that extend too far west compared to

observations (Vaittinada Ayar et al., 2023).

Sub-seasonal variability exhibits the smallest spatial scales for both SChl and SST (Figure 5 a,c). In satellite-derived SChl, sub-seasonal variability has spatial scales of around 100-150 km, at the resolution limit of the grid to which we regridded the satellite data (100 km). Using the same satellite product at 25 km resolution, Keerthi et al (2022) identified sub-seasonal spatial

scales of around 50 km. Most simulated sub-seasonal variability of SChl, except in MPI-ESM1.2-HR, is characterized by spatial scales exceeding 350 km. Among the simulations, the CESM2 models exhibit the largest scales close to 500 km, followed by the MPI models (excluding MPI-ESM1.2-HR), then IPSL, CNRM, and NorESM2 models. MPI-ESM1.2-HR, an eddy-permitting model, exhibits the smallest scales (~200 km) for sub-seasonal timescales, although still larger than in observations. The mean spatial scale corresponding to SST sub-seasonal variability is ~200 km in observations. In contrast,

simulations, with the exception of MPI-ESM1.2-HR, display scales exceeding 600 km.

**3.4 Drivers of sub-seasonal variability**

Sub-seasonal SChl fluctuations result from various drivers across different spatial scales: submesoscale/mesoscale processes (1–100 km), cyclones and tropical storms (100–1,000 km), large-scale climate modes (>1,000 km) and internal variability. A prior study (Keerthi et al., 2022), based on satellite ocean color SChl measurements, suggests that sub-seasonal variability is

strongly associated with mesoscale and submesoscale variations, as evidenced by their mean spatial scales of about 50 km. However, in the high latitudes and tropics where sub-seasonal variability has a large contribution to the SChl total variability, >50% of the sub-seasonal variability has spatial scales greater than 100 km. At high latitudes, these large spatial scales reflect synoptic storms (Prend et al., 2022; Keerthi et al., 2022; Thomalla et al., 2011), while in the tropics, they reflect intraseasonal climate modes such as Madden Julian Oscillations, Kelvin waves, and tropical instability waves (Keerthi et al., 2022;

Resplandy et al., 2009; Strutton et al., 2001; Xu et al., 2018; Jin et al., 2013).

We observe a significant reduction in the relative contribution of sub-seasonal variability to the total SChl variance in the satellite SChl product compared to that in Keerthi et al., (2022). This disparity can be attributed to the spatial regridding process used to analyse the satellite SChl data at a mean horizontal resolution comparable to that of the CMIP6 models. In the present

study, the horizontal resolution of the satellite product is approximately 100 km, while in Keerthi et al., (2022), it is 25 km. Since the mean spatial scale of sub-seasonal variability has been shown to be of the order of 50 km (Keerthi et al., 2022),



regridding to coarser resolution removes substantial variability. Due to effective coarse-resolution, which is several times lower than grid resolution (Levy et al., 2012), CMIP6 simulations should structurally underestimate sub-seasonal variability. However, contrary to this expectation, most models overestimate the relative contribution of sub-seasonal variability to the total SChl variance. Furthermore, the mean spatial scale is about 4 to 6 times larger than in observations.

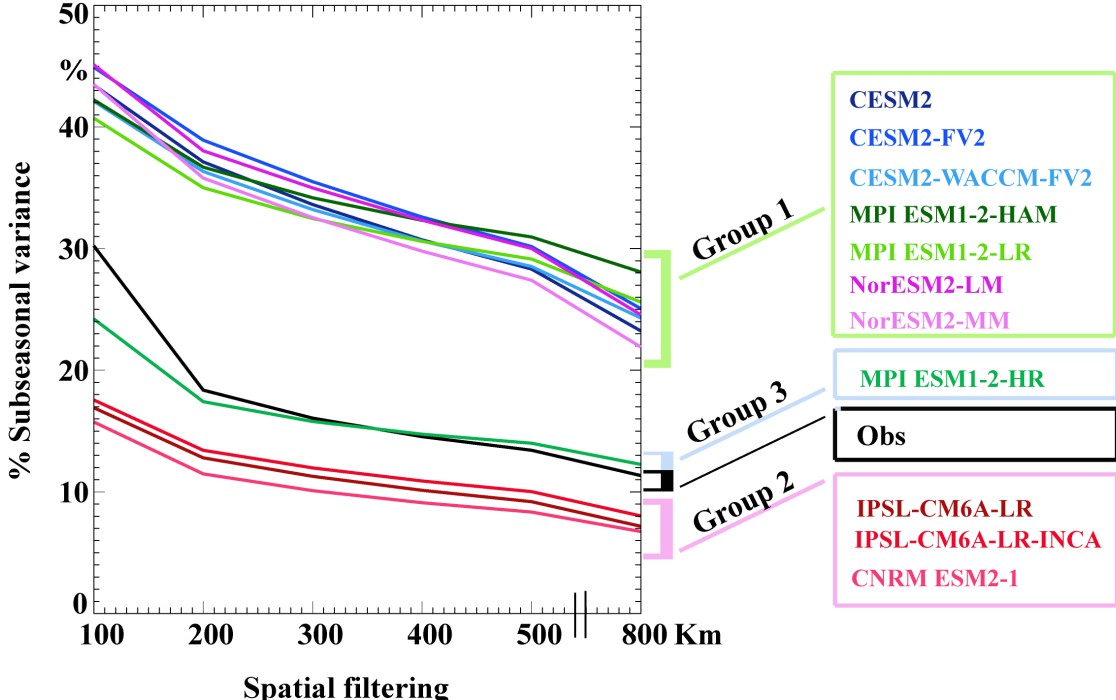

**Figure 6: Sub-seasonal variance across spatial scales:** Percentage of total SChl variance explained by sub-seasonal variability after applying a spatial smoothing of varying scale from 100km to 800Km to the SChl data. Coloured boxes for the model names represent the different model groups described in Section 3.2, with green for Group 1, pink for Group 2, and blue for Group 3.

When the sub-seasonal variance is assessed at spatial scales from 100 to 800 km, a clear pattern emerges in the SChl data. The relative contribution of sub-seasonal variations to total SChl variance in satellite observations drops from 30% at 100 km to 18% at 200 km, followed by a gradual decline up to 800 km (Figure 6). Using the categories of models defined in section 3.2, the model in category 3 (MPI-ESM1.2-HR) shows a decline of 7% from 100 km to 200 km to values relatively close to the observations. Above 200 km, the simulated sub-seasonal relative contribution then decreases similar to observations. Models in category 2 also exhibit a pattern similar to that of observations with a modest decrease of (~5%) from 100-200km and then a gradual decrease up to 800km. But these models underestimate the contribution of sub-seasonal variability by about 4-5% at all spatial scales from 200-800km. This suggests that these models correctly simulate the large-scale component of sub-seasonal variability but are unable to capture its small-scale component as expected due to their limited resolution. Category





1 models systematically overestimate the sub-seasonal contribution by about 20%. In addition, the downward slope tends to be steeper than observed, particularly in CESM2 configurations. We hypothesize that the large sub-seasonal variability simulated by models in this category is generated by driving mechanisms that differ from observations.

**Figure 7: Sub-seasonal SChl variability across temporal subperiods:** Pie diagram showing the relative contribution of each time period to the total SChl sub-seasonal variance in the observations and different CMIP6 historical simulations. Coloured boxes represent the different model groups described in Section 3.2, with green for Group 1, pink for Group 2, and blue for Group 3.





The discrepancies between the three model categories in capturing SChl sub-seasonal variability are also evident in an analysis of temporal subscales (Figure 7). Satellite observations show that 40-50% of sub-seasonal variability is present in the 16-32day temporal window, followed by 20% in the 33-48day window, and 0-15% in days 49-64. Category 2 and 3 models show a similar pattern, but tend to underestimate the contribution of the 16-32day period, probably due to an underestimation of mesoscale variability. In contrast, all other simulations predominantly overestimate sub-seasonal variability in the 16-32day

period.

Much of the phytoplankton variability is often attributed to fluctuations in their physical environment. However, phytoplankton time series also exhibit variability that is not strongly correlated with key physical variables and is distinctly nonlinear (Mayersohn et al., 2021, 2022). These intrinsic oscillations are primarily associated with two mechanisms: one related to

species succession and resource changes (Tilman, 1977; Huisman and Weissing, 1999, 2001), and the other to changes in total phytoplankton biomass due to predator-prey interactions (Gilpin, 1979; Edwards and Brindley, 1996). Predator-prey oscillations typically occur on shorter subseasonal time scales of up to 60 days while resource-related oscillations can extend to longer time scales (Mayersohn et al., 2021, 2022). This intrinsic variability can complicate the accurate simulation of high-frequency, subseasonal fluctuations in phytoplankton populations by models.


A recent study (Rohr et al., 2023) emphasizes significant differences in CMIP6 simulations regarding the representation of zooplankton-specific grazing, which could have a profound impact on the temporal variability of phytoplankton (Mayersohn et al., 2021, 2022; Talmy et al., 2024). Despite the diversity of functional roles and distributions of zooplankton species (Kiorboe, 2011; Benedetti et al., 2023; Pata and Hunt, 2023), biogeochemical (BGC) models must represent aggregated

behaviour using a limited number of zooplankton groups. This limitation introduces considerable uncertainty into the modeling of complex zooplankton communities and their role in the marine carbon cycle. The representation of grazing in CMIP-class BGC models varies considerably, from models with a single zooplankton functional type grazing on a specific phytoplankton type to those incorporating multiple zooplankton groups and potential preys (Sailley et al., 2013; Petrik et al., 2022; Rohr et al., 2023). Beyond differences in grazing formulations, there are significant variations between models in terms of

phytoplankton groups/size classes, temperature-dependent phytoplankton growth, biogeochemical factors influencing phytoplankton growth, and resource competition (Séférian et al., 2020), further contributing to the overall uncertainty in model simulations. For instance, many studies have shown that the mathematical formulation and the parameter values of the closure term representing predation by unresolved higher trophic levels have profound impacts on the temporal stability of the biogeochemical model, especially at high resource levels (e.g., Edwards and Brindley, 1999; Edwards and Yool, 2000; Omta

et al., 2023).



Analysing the relative significance of each factor—biogeochemical models, ocean dynamical models, and horizontal resolution—influencing the observed differences between simulations in capturing the temporal variability of SChl proves challenging. For example, even though MPI-ESM1.2-HR and MPI-ESM1.2-LR use similar ocean, atmosphere, and
biogeochemical models, they show notable differences in the simulation of SChl temporal variability, with improved horizontal resolution playing a crucial role. Interestingly, models like IPSL and CNRM, classified as type 2, which differ from MPI-ESM1.2-HR in ocean, atmosphere, and biogeochemical components and utilize a coarse horizontal resolution (100 km), show comparable, if not superior, performance in simulating SChl temporal variability. This is despite a minor underestimation in sub-seasonal variability contribution to the total SChl variance. Furthermore, the MPI and NorESM models, which share the
same biogeochemical model but not the same physical ocean model, simulate contrasting SChl variability.

**3.5 Role of sub-seasonal variability in year-to-year variations of SChl**

Recent studies have highlighted the significant role of sub-seasonal variability in modulating annual variations in SChl (Keerthi et al., 2022; Prend et al., 2022; Levy et al., 2024). In the Southern Ocean and western and eastern boundary upwelling systems, high frequency sub-seasonal SChl variations can accumulate and modulate the year-to-year variations in SChl. We quantify
the impact of sub-seasonal variability on year-to-year SChl variations using the annual mean low-frequency index, following the methodology of Keerthi et al., (2022). This index is computed as the squared correlation between the annual mean SChl and the mean multi-annual component of SChl for each year. An index approaching 1 implies that year-to-year variations are primarily associated with multi-annual variations, while an index less than 1 indicates a contribution of sub-seasonal and seasonal variability to year-to-year variations in SChl. A smaller index thus signifies a greater contribution of sub-seasonal
variability to year-to-year variations.

For satellite-derived SChl data with a horizontal resolution of 25km, consistent with Keerthi et al. (2022), the index is very close to 1 in parts of the tropics and subtropics. However, it decreases, reaching values as low as 0.5, in regions such as the equatorial Atlantic, the Southern Ocean, western boundary current regions and eastern boundary upwelling systems, where
sub-seasonal variability and irregular seasonal cycles intensify. These regions are characterized by substantial eddy and frontal activity. When regridded to a 100km horizontal resolution, the index is close to 1 over most of the ocean, except in the southern subpolar regions and in the northern Indian Ocean (Figure 8a). This implies that much of the sub-seasonal variability that imprints on year-to-year variations is due to mesoscale processes.

The distribution of the index computed for the CMIP6 models is presented in Figure 8 b-l. Models simulate contrasting distributions which can be structured according to the three categories defined in section 3.2. Category 2 models, with the exception of CNRM ESM2-1, show sporadic contributions in the Southern Ocean. All models in category 1 strongly overestimate the impact of sub-seasonal variability on year-to-year variations in the Southern Ocean. The category 3 model





shows a similar pattern to observations with a slight overestimation. This is further supported by regression of annual mean

SChl with the annually-averaged sub-seasonal component of SChl (Supplementary Figure 1). In the observations, sub-seasonal variability contributes 10 to 30% of the amplitude of year-to-year variations in the Southern Ocean. In models, with the exception of category 2 models, this contribution is significantly larger, exceeding 30% in large parts of the Southern Ocean.

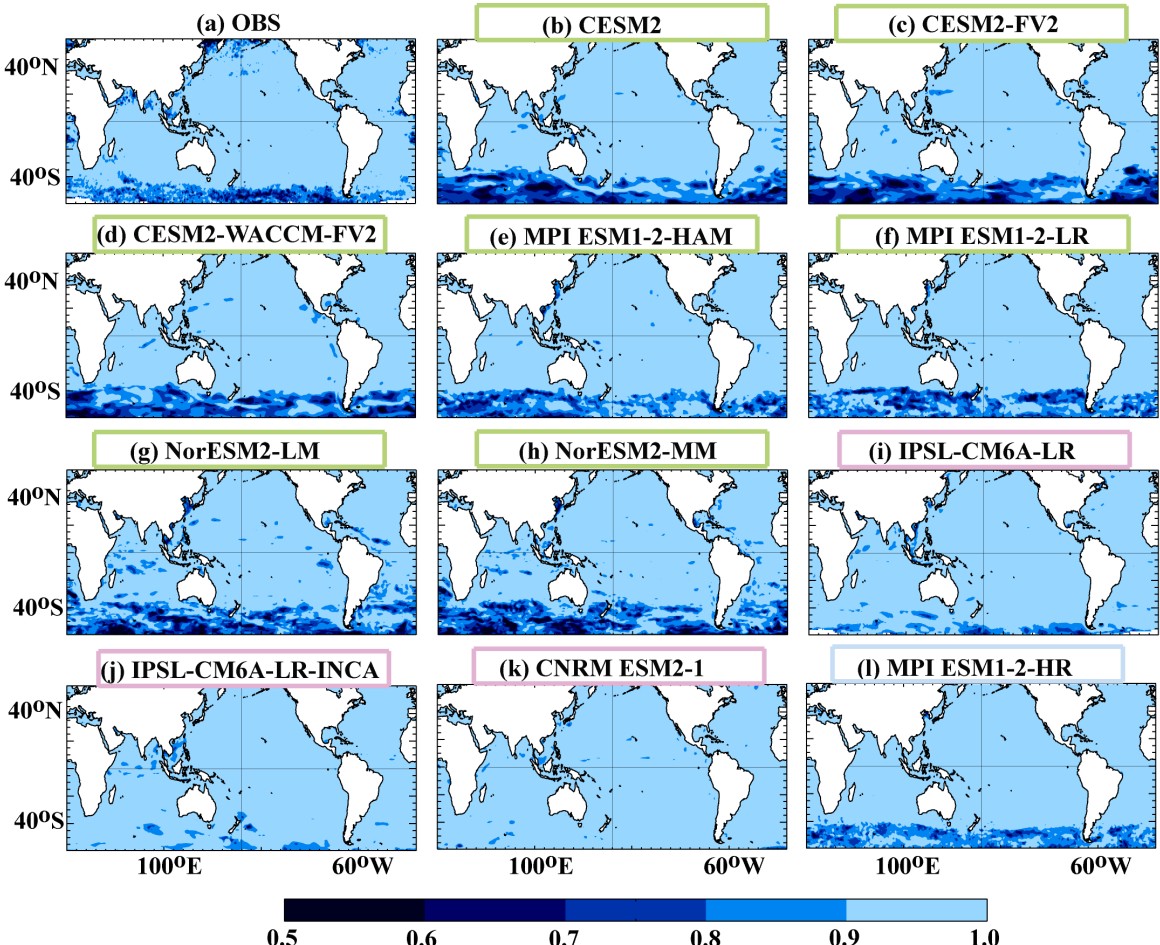

**Figure 8: Annual mean low-frequency index for SChl**, which is defined as the correlation square between annual mean and annual mean of the multi-annual component. When the index is close to one, year-to-year fluctuations in the annual mean reflect low-frequency variability. The value of the index decreases as high-frequency variability contributes more to year-to-year variations. Coloured boxes for the model names represent the different model groups described in Section 3.2, with green for Group 1, pink for Group 2, and blue for Group 3.



**3.6 Projected changes to SChl temporal variability**

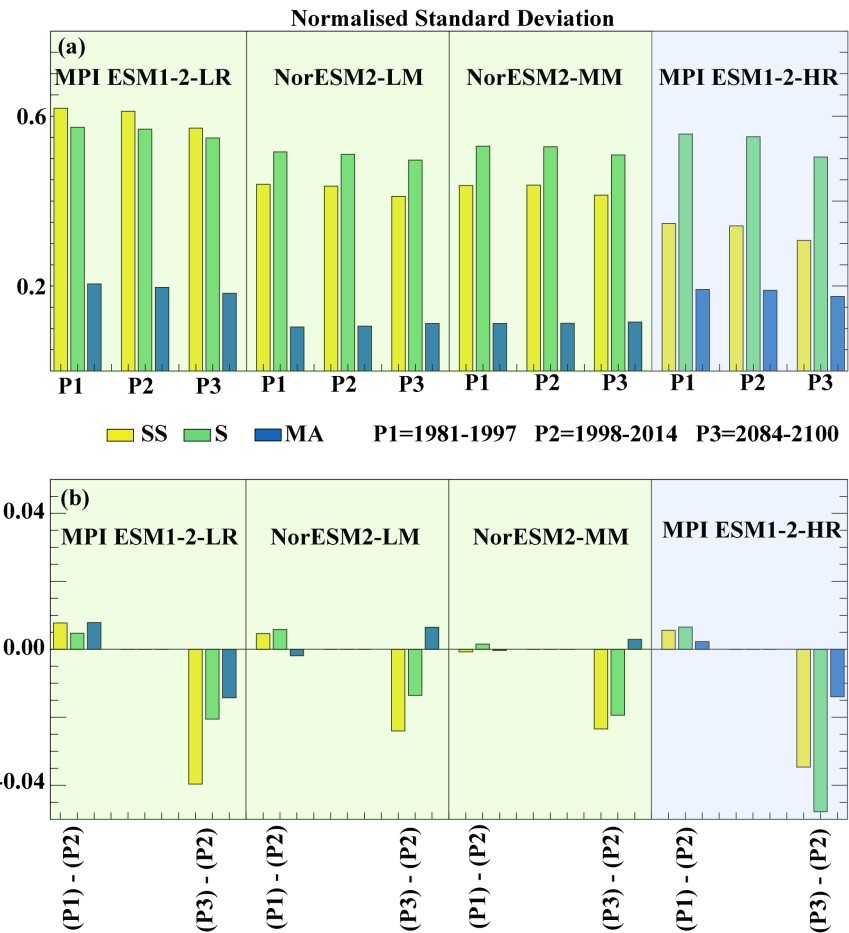

**Figure 9: Future projections** (a) Normalised standard deviation of SChl from two different periods of CMIP6 historical simulations (1981-1997 & 1998-2014) and CMIP6 SSP585 simulation for the period (2084-2100). (b) Difference between
different periods considered. Shading in panel a and b represents the different model groups described in Section 3.2, with green for Group 1, and blue for Group 3.

Under climate change scenarios, ESMs consistently project increased stratification and a reduction in nutrient concentrations in the euphotic zone (Bopp et al., 2001; Sarmiento et al., 2004; Cabré et al., 2014; Fu et al., 2016). These changes generally
lead to an overall reduction in net primary production due to increased limitation of phytoplankton growth by nutrients (Bopp et al., 2013; Krumhardt et al., 2017; Moore et al., 2018), although results from CMIP6 models show a more modest reduction associated with greater uncertainty than CMIP5 models (Kwiatkowski et al., 2020; Tagliabue et al., 2021). In addition to these simulated mean changes, global warming has been also shown to alter seasonal cycles (Henson et al., 2013; Thomalla et al.,



2023), to modify interannual and decadal climate modes (Cai et al., 2014, 2021), and to increase the frequency of extreme
events like heatwaves and tropical cyclones (Frolicher et al., 2018; Knutson et al., 2020; Walsh et al., 2016; Jo et al., 2022).
The multi-model mean seasonal amplitude of global SST is projected to increase by +0.59±0.21°C under SSP5-8.5
(Kwiatkowski et al., 2020), mainly resulting from an overall shoaling and increasing seasonal amplitude of the mixed layer
(Alexander et al., 2018; Jo et al., 2022). By the end of the 21st century, most models forecast an increase in frequency and
amplitude of central Pacific El Niño events and a rise in the frequency of eastern Pacific El Niño events (Vaittinada Ayar et
al., 2023). The increased frequency of extreme events such as marine heatwaves (Frolicher et al., 2018) and tropical cyclones
(Knutson et al., 2020; Walsh et al., 2016), coupled with mesoscale and submesoscale variability linked to global warming
scenarios (Martínez-Moreno et al., 2021, 2022), contributes to an increase in sub-seasonal variability. With respect to the SChl
temporal variability, our knowledge of its potential changes is more limited. To our knowledge, an analysis of the simulated
change of the sub-seasonal variability of SChl, using for instance CMIP-type models is lacking.

Here, we analyzed the changes in SChl temporal variability under the high-warming scenario SSP5-8.5, using simulations that
provide daily SChl outputs to be able to include sub-seasonal variability in our analysis. SChl standard deviations at sub-
seasonal, seasonal and multi-annual timescales for the end of the century (2084–2100) and for the end of historical simulations
(1998-2014) are compared in Figure 9. To determine whether the changes noted between the two periods can be explained by
decadal variability, we also applied our analysis to the 1981-1997 period. All models simulate a consistent decrease in the
normalized standard deviation of both seasonal and sub-seasonal timescales from over the 21$^{st}$ century. The MPI models tend
to simulate a stronger decrease at both timescales (7-10% sub-seasonally and about 3-8% seasonally), with larger changes in
the high resolution configuration. In NorESM models the simulated decrease is smaller, 4-5% and 2-3% at sub-seasonal and
seasonal timescales respectively. At multi-annual timescales, changes are of a similar relative magnitude, about 3-8% but show
opposite signs between MPI and NorESM models. Comparison with the period 1981-1997 and a similar analysis carried out
with the piControl experiments (Figure S2) shows that these changes cannot be explained by decadal natural variability.

## 4 Summary

In this study, we assessed how ESMs participating in CMIP6 reproduce surface phytoplankton variations across various
temporal scales, with a particular focus on the often-overlooked sub-seasonal timescales. We compared 13 ESMs that have
daily SChl outputs for the historical period (1998-2014) with the ESA OC-CCI merged ocean color satellite SChl product.
Unlike SST, where ESMs generally exhibit consistent behaviour, we find significant intermodel variability and discrepancies
between SChl simulations and observational data, both in terms of the amplitude of variability and likely driving mechanisms.
Our findings indicate that none of the analyzed models accurately replicate both the observed variability across timescales and
their relative contributions to the total temporal variance in SChl.





Based on globally averaged metrics of sub-seasonal timescales we categorized the models into three distinct groups. Group 1
models strongly overestimate sub-seasonal SChl standard deviation and its relative contribution to total variance, despite their
coarser horizontal resolution compared to observations. Group 2 models better represent the observed SChl variability across
timescales but underestimate the sub-seasonal variance and its relative contribution to total variance. These models capture
large-scale sub-seasonal variability but fail to resolve small-scale components, partly due to their resolution, which does not
permit mesoscale processes—a bias that can be reduced with higher-resolution models. Group 3 models correctly simulate the
relative contribution of different timescales to total variance but significantly overestimate SChl variances. This overestimation
of sub-seasonal variance in Group 1 and Group 3 models is possibly due to intrinsic oscillations (e.g., predator-prey
oscillations) inconsistent with observations and potentially stemming from the structure of the biogeochemical models. Models
that overestimate sub-seasonal variability exaggerate its influence on annual variations, potentially impacting long-term trends.
In contrast, Group 2 models exhibit a diminished impact of sub-seasonal variations on annual variations, which could also
influence long-term projections.

Overall, our findings highlight the challenges and discrepancies in ESM representation of surface phytoplankton dynamics,
emphasizing the crucial role of spatial resolution and the accurate representation of biogeochemical processes in determining
model accuracy. A direct relationship between model performance and horizontal resolution might not always exist. Group 2
models, despite having a lower resolution comparable to Group 1 models, exhibit comparatively better performance among
the models analyzed. However, increasing the resolution of both the atmospheric and ocean components of a model
significantly improved its performance.

By the end of the 21st century, models project a modest global decrease in both seasonal and sub-seasonal variability. However,
projected changes on multi-annual timescales diverge. This analysis is however limited by the number of models that provide
daily output for the historical and future periods (only four ESMs). We therefore advocate that, in future exercises, more
modeling groups submit daily surface outputs of biogeochemical variables, particularly, but not only, SChl. The poor capability
of the models at simulating sub-seasonal SChl dynamics casts doubt on projections at these temporal scales and potentially
also limits long-term projections due to the ability of sub-seasonal dynamics to influence year-to-year variations.

**Acknowledgments**

The authors acknowledge the support from the ENS-PSL CHANEL chair and the Centre National d'Etudes Spatiales
(TOSCA).



**Author Contributions**

MGK, OA, ML and LK conceived and developed the study. MGK performed the data analysis and made the plots. MGK, OA, LK and ML made the interpretation of the results and wrote the manuscript.

**Data Availability**

All data analyzed in this study are freely available from the respective websites mentioned in the Methods Section.

**Competing interests' statement:**

The authors declare no competing interests

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
