# Peer review of "Inadequacies in the representation of sub-seasonal phytoplankton dynamics in Earth system models"

_EGUsphere, 2024_

## Author Comment (AC1)

We thank the Reviewer for their positive feedback and valuable inputs on the manuscript. Below, we provide a detailed response to each of the comments — reviewer comments are in black, and our responses are in blue.

**Summary:**

The manuscript makes use of the subset of models from CMIP6 that provided high temporal resolution SST / chlorophyll output to investigate the sub-seasonal dynamics of simulated phytoplankton. The model output is processed first using a decomposition methodology that breaks down the variability into different temporal modes. Subsequently, the output is further processed using spatial decomposition to identify whether the simulated variability has the correct horizontal length scales (e.g. to distinguish where models exhibit seemingly comparable variability to observations but on much coarser spatial resolution). The manuscript finds that none of the models realistically represents the seasonal and sub-seasonal patterns of variability observed (unlike the situation with SST). However, the spatial decomposition teases out patterns between models that allows them to be separated into three groups with better or worse representation of real-world variability. The authors note that one group is hampered by its spatial resolution, while the other two exhibit excessive sub-seasonal variability apparently from tightly-coupled predator-prey cycles. The manuscript concludes by noting the value of high temporal resolution output for identifying unrealistic model behaviour, and with a call for modelling groups to endeavour to provide this in CMIP7.

**Review:**

Overall, I found the manuscript interesting and quite convincing about the realism or otherwise of current generation CMIP models. I have no major comments on the content of the manuscript, but have made a small number of minor suggestions about improvements. I recommend accepting the manuscript following these minor corrections.

We thank the reviewer for their encouraging assessment of our manuscript.

**Comments:**

One overall comment I have is around the quality of the figures. There are some unhelpful choices here to my mind and I detail these below. However, I would accept that this is largely an aesthetic decision, and would not insist on my suggested changes being implemented. Another general comment I'd make is that it would be good to try to put the models examined into some sort of context within the wider CMIP6 ensemble – I've suggested an idea in what I say about Figure 2a, but there may be a more obvious or better solution.

Ln. 46: Inconsistent ordering of references; they're neither in chronological nor alphabetical order (I prefer the former).

The references are now arranged correctly – 'This is particularly critical for phytoplankton as it is characterized by large natural variability at diverse timescales, which often masks the long-term trends (Henson et al., 2010, 2016; Doney et al., 2014; Keerthi et al., 2022)'.

Ln. 64: Amend to "… produced by *a subset of* ESMs …".

The sentence is corrected – 'Capitalizing on high frequency global measurements of satellite ocean color SChl, we evaluated the performance of historical simulations produced by a subset of ESMs participating in the Coupled Model Intercomparison Project Phase 6 (CMIP6) to simulate global surface ocean phytoplankton dynamics across diverse temporal scales (sub-seasonal, seasonal, and multi-annual), with a specific focus on high frequency sub-seasonal variability'.

Table 1: The IPSL and CNRM models are lumped together (presumably because of a common ocean), but do they share a common atmosphere or atmospheric resolution?

The IPSL and CNRM models are grouped together because they share the same ocean model. However, they utilize different atmospheric models: **ARPEGE - Climat** for the CNRM models and **LMDZ** for the IPSL models. They are also based on different land surface models as well as different sea-ice components. All models are grouped based on their shared physical and biogeochemical (BGC) ocean components.

Table 1: The MPI rows have a missing border between HAMOCC6 and 150 km cells.

This has been corrected.

| CMIP6 Simulations | Physical Ocean Model | Ocean BGC Model | Horizontal resolution (Physical & BGC Model) | Model Simulations | References |
|---|---|---|---|---|---|
| IPSL-CM6A-LR | NEMO-OPA | PISCES | 100 km | Historical | Boucher et al. 2018, 2021; Séférian, 2018 |
| IPSL-CM6A-LR-INCA | | | | | |
| CNRM-ESM2-1 | | | | | |
| CESM2 | POP2 | MARBL | 100 km | Historical | Danabasoglu, 2019a, b, c; |
| CESM2-FV2 | | | | | |
| CESM2-WACCM-FV2 | | | | | |
| MPI-ESM1.2-HAM | MPIOM | HAMOCC6 | 150 km | Historical | Neubauer et al., 2019 ; Wieners et al., 2019a, b, c; Jungclaus et al., 2019a, b; Schupfner et al., 2019 |
| MPI-ESM1.2-LR | | | | Historical, SSP5-8.5, piControl | |
| MPI-ESM1.2-HR | | | 40 km | | |
| NorESM2-LM | MICOM | HAMOCC | 100 km | Historical, SSP5-8.5 piControl | Seland et al., 2019a, b, c ; Bentsen et al., 2019a, b |
| NorESM2-MM | | | | Historical, SSP5-8.5 | |

**Table 1.** The CMIP6 Earth system models used in this study, their physical and biogeochemical ocean components, nominal horizontal ocean resolution and the simulations assessed.

Table 1: The MPI and NorESM2 rows mention piControl simulations, but I don't believe that these are mentioned elsewhere in the manuscript.

PiControl simulations are shown in Supplementary Figure 2 (Figure S2) and are described in line 485.

Ln. 134: A period of 8 months is mentioned here for the so-called "multi-annual component". Why 8 months and not 12 months? I'm sure I'm not understanding something.

The multiannual component is defined as low-frequency variability characterized by timescales of approximately one year or longer. We did not impose a strict cut off at 12 months, thus this component encompasses variability with periodicities extending beyond 8 months. However, when analysing time series at specific locations, we observed that the signal within the 8–12-month range was relatively weak, suggesting that the dominant contributions to the multiannual variability arise from longer timescales. This flexible approach ensures a more inclusive representation of low-frequency variability without being constrained by rigid temporal boundaries. For more information on the temporal decomposition we applied here, please refer to Keerthi et al., 2020.

Keerthi, M. G., Levy, M., Aumont, O., Lengaigne, M. & Antoine, D.: Contrasted contribution of intraseasonal timescales to surface chlorophyll variations in a bloom and an oligotrophic regime. Journal of Geophysical Research: Oceans, 125(5), e2019JC015701, 2020.

Ln. 153: I'm not a fan of "Results and Discussions" sections, and would prefer the authors to properly separate results from discussion to improve the manuscript's clarity. However, it can be difficult to separate them at this stage, so ignore this suggestion if it isn't obvious to address.

In the initial stages of the manuscript, we attempted to present the results and discussion in separate sections. However, this approach led to some repetition of content. Consequently, we decided to combine them into a single "Results and Discussion" section. At this stage, separating them would be overly burdensome. Thank you for your understanding.

Figure 1: Conventionally, darker colours are used to indicate lower values while brighter colours are used to indicate higher values. The choice here is confusingly the reverse.

We have updated the figure, using a color scheme where darker tones represent lower values and brighter tones indicate higher values. The revised figure can replace Figure 1 in the manuscript.

[Figure]

Figure 1: Mean state evaluation: Annual mean SChl (a) Observed (ESA OC-CCI product) and (b) CMIP6 multi-model mean for the years 1998-2014 and domain 60ºN-60ºS. (c & d) Similarly for SST.

Figure 2a: The models are distributed into two clear groups but the manuscript doesn't reflect on this. Is there any straightforward distinction to be drawn between them? For instance, what would the mean fields of the two groups look like? Would there be any clear distinguishing patterns.

MPI models exhibit significant overestimation in both mean field and variance across all temporal scales. A comparative figure showing the mean field of the MPI models against other models is attached. This figure shows that spatial patterns are similar but MPI models consistently overestimate the magnitude across the global ocean.

[Figure]

Figure 2: CMIP6 mean state evaluation: a) Ensemble mean SChl from the IPSL, CNRM, CESM, and NorESM models analyzed in this study; b) Ensemble mean SChl from the MPI models analyzed in this study.

Figure 2a: Since the models examined fall into only 4 "families", and given that they all perform fairly badly here, I wonder if it might be worthwhile somehow contextualising their performance against the wider CMIP6 ensemble? Possibly by adding other models that are

outside of the analysis here? Either in this figure, or in a supplementary version of this figure. Even without those models being analysed in detail as here, it would provide context for the representativeness of the models used here.

In this manuscript, we analyzed the SChl temporal variability simulated by the models, with a particular focus on the subseasonal timescale, which is often overlooked. So we included only models that provide SChl data at daily temporal resolution, necessary to properly evaluate subseasonal signals.

A more detailed analysis of the mean chlorophyll surface distribution predicted by a larger set of CMIP6 models is provided in Séférian et al. (2020). Figure 2 of this study displays the model-data deviations for this larger set of the same period (1998-2014) as the one used in our study.

Séférian et al.: Tracking improvement in simulated marine biogeochemistry between CMIP5 and CMIP6, Current Climate Change Reports, doi:10.1007/s40641-020-00160-0, 2020.

Figure 3: This chart makes a sensible comparison between the variability modes of the obs and models. However, I wonder if there's a way to put the information it presents onto a single axis where the models and observations can be seen together. For instance, "total" variability on the x-axis, and the fraction that's sub-seasonal on the y-axis? You may have tried something like this already.

Thank you for your suggestion. We did explore several alternative visualization methods. However, after multiple iterations, we found that the current figure is the most effective way to clearly and accurately convey the variability modes.

Figure 4: Add in the caption which models, and why, are missing here. Presumably data availability?

The caption has been modified to – '**Figure 4: SST variability across timescales.** This figure is similar to Figure 3 but focuses on SST. Note that NorESM2-LM and NorESM2-MM are excluded, as daily SST data for these models is not available on the CMIP6 data portal.'

Figures 4, 5: A bit more consistency in style would be good for these bar chart figures. Figure 3 seems to make use of the space best, with Figures 4 and, especially, 5 using it less well (i.e. the bars are thinner).

Figures 3 and 4 are of the same size and have the same bar widths. The size discrepancy occurred when pasting the TIFF files into the word document. In Figure 5, we attempted to include both SChl and SST, as the information provided by these figures is less detailed than that in Figures 3 and 4.

Figure 7: I think this could be a much better figure if pie charts weren't used. Each model (and possibly model region) could be given a simple x-y subplot in which the x-axis is period and the y-axis is geographical area or frequency. Each subplot could then also contain the same information for the observational data. This would add information currently hidden by the limited number of periods selected for the pie charts, and would make it easier to compare with the observational data. At present the reader has the unenviable task of squinting to try to work

out how similar / different one pie chart is from another. Line plots would – I suggest – be much better here.

Thank you for this suggestion. We agree with the reviewer that the pie charts can be difficult to compare. As a result, we have changed it into a bar diagram for, we hope, better clarity and comparison.

[Figure]

Figure 7: Sub-seasonal SChl variability across temporal subperiods: Bar plot showing the relative contribution of each temporal period to the total SChl sub-seasonal variance in the observations and different CMIP6 historical simulations.

This bar diagram can replace Figure 7 in the manuscript.

Figure 8: This is a horrible colour map. Not only is it a single colour, but the different shades of that colour are very difficult to discern, with an emphasis on darker shades that make any distinctions in the plots fairly invisible. Why not use one of the colour maps used elsewhere to make discerning the distinctions easier?

We utilized the same color scale as in Keerthi et al. (2022) to facilitate direct comparison between the results of this study and those presented by Keerthi et al. (2022).

Keerthi MG, Prend CJ, Aumont O, Levy M.: Annual variations in phytoplankton biomass driven by small-scale physical processes. Nature Geoscience :1–14, 2022.

Ln. 502: The structure of biogeochemical models is alluded to here but no evidence is presented. Perhaps illustrating with time-series plots of representative differences between models might help clarify this here. Or even examine the low frequency output of the models involved to determine if they differ in their phytoplankton-zooplankton relationships. However, this is only a suggestion as it might be sending you on a wild goose chase.

Thank you for your suggestion. We agree that illustrating the differences between models using time-series plots or examining the low-frequency output to assess differences in phytoplankton-zooplankton relationships could provide valuable insights. However, this is a challenging aspect to address, and adding this analysis would significantly extend the manuscript. We refer to Rohr et al. (2023), which discusses the largest source of inter-model uncertainty in marine biogeochemical models, specifically regarding phytoplankton-specific loss rates to zooplankton grazing. Rohr et al. (2023) found that this uncertainty is more than three times larger than that of net primary production and is driven by large differences in prescribed zooplankton grazing dynamics. Given these findings, further exploration of phytoplankton-zooplankton interactions across the models may indeed provide a deeper understanding, but we feel this would require substantial work beyond the scope of the current manuscript.

A likely better way to study the impacts of coupling assumptions between phytoplankton and zooplankton would be to use a single modeling framework to explore the major differences highlighted by Rohr et al. (2023). Otherwise, the many differences in the representation of marine biogeochemistry and other components of Earth System Models, would almost certainly prevent attribution.

Ln. 515: The authors advocate for CMIP modelling groups to submit daily outputs of biogeochemistry variables but don't mention which ones specifically. Obviously chlorophyll but, per the preceding point, would they advocate for others like surface zooplankton too? This is a good opportunity to advocate for them.

According to Rohr et al. (2023), there are significant differences in the prescribed zooplankton grazing dynamics among CMIP6 simulations, which leads to considerable variations at higher frequency timescales. In particular, predatory-prey oscillations are suspected here but proved to be extremely difficult to evidence without corresponding zooplankton information. Having this information, i.e. zooplankton concentrations and grazing rates at daily resolution would be very useful in this regard.

---

## Author Comment (AC2)

We thank the Reviewer for their positive feedback and valuable inputs on the manuscript. Below, we provide a detailed response to each of the comments — reviewer comments are in black, and our responses are in blue.

**General comments:**

The writing is clear and the manuscript introduces new findings that contribute to our understanding of chlorophyll-a dynamics. I recommend accepting the manuscript with minor revisions. The following comments are merely provided as suggestions to further improve the manuscript's completeness and clarity.

We thank the reviewer for their encouraging assessment of our manuscript.

**Summary:**

This study compares simulated surface chlorophyll-a (Schla) variability from a subset of CMIP6 Earth System Models (ESMs) with satellite observations and contrasts this performance with that of SST. The analyses highlight discrepancies in the ability of ESMs to simulate Schla across different timescales, with a specific focus on the understudied sub-seasonal variability. The ESM simulations are selected based on the availability of daily Schla and SST outputs. Temporal variability is decomposed into sub-seasonal, seasonal, and multi-annual scales, identifying three main groups: one showing an overestimation of sub-seasonal variability which is attributed to the coarse spatial resolution of the ESMs, a second group showing an underestimation of sub-seasonal variability, potentially linked to intrinsic predator-prey oscillations within the ESMs, and a third group displaying an overestimation of total variance but consistent temporal decomposition. The authors conclude that, unlike SST, ESMs do not adequately represent Schla variability, emphasizing the need for additional CMIP simulations with higher spatial and temporal resolutions to address these limitations.

**Specific comments:**

Perhaps the manuscript could include an explicit mention of the limitations of the approaches and how they affect the final findings, specifically concerning:

- The biases/uncertainties of satellite observations: While the manuscript uses satellite observations as the benchmark for comparison, it would strengthen the discussion to acknowledge the inherent biases and limitations of these datasets. For instance, biases introduced by gap-filling and uncertainties in the retrieval process could affect the representation of SChla variability. It would be beneficial if the authors discussed these biases and how they might influence the overall findings.

We will add a new paragraph highlighting the biases and uncertainties in satellite observations in line 275: "It should be noted that biases introduced by gap-filling in satellite-derived data can lead to an inaccurate representation of SChl variability, as missing or interpolated data points may not capture the true temporal or spatial patterns of chlorophyll concentrations. Additionally, uncertainties in the retrieval process, such as atmospheric corrections and sensor calibration, can further distort the observed variability, affecting the reliability of satellite-derived estimates of surface chlorophyll. However, satellite ocean color measurements remain the only available source of high-frequency observations of SChl over extended periods at a global scale. Furthermore, a comparison of SChl at a mooring location in the BOUSSOLE in

the Gulf of Lion showed that satellites can capture SChl variability at higher temporal resolutions reasonably well (Keerthi et al., 2020). Nonetheless, cloud cover remains a limitation that can affect the accuracy of these measurements."

Keerthi, M. G., Levy, M., Aumont, O., Lengaigne, M. & Antoine, D.: Contrasted contribution of intraseasonal time scales to surface chlorophyll variations in a bloom and an oligotrophic regime. Journal of Geophysical Research: Oceans, 125(5), e2019JC015701, 2020.

- Comparison of satellite and ESM timeseries: The analyses use satellite timeseries spanning 16 years and ESM simulations spanning 33 years. It would be helpful if the authors addressed whether this difference could impact the representation of multi-annual variability in the analyses and thereby affect their findings and conclusions.

This was previously insufficiently clear. We provided the comparison between satellite SChl and historical CMIP6 simulations for the common period of 1998–2014. To clarify this, we will revise line 110 by adding the sentence: 'The comparison between satellite observations and CMIP6 historical simulations is provided for the common period, 1998–2014.'

The CMIP6 data for 1981–1997 is used exclusively for Figure 9, to explore whether the changes observed between the two periods (1998-2014 and 2084-2100) can be attributed to decadal variability.

- Thresholds for spatial coherence analysis: When mentioning the thresholds for the analyses of the spatial extent of coherence, the authors could clarify the rationale behind the choice for an upper threshold of 2400 km in diameter and the threshold value of 0.8 for correlations and how these influence the findings and interpretations.

Our focus was on the spatial scales of subseasonal and multiannual variability, which typically occur at smaller scales, predominantly below 2000 km. Setting an upper threshold of 2400 km allowed us to concentrate on the relevant scales while reducing computational time and energy consumption.

The choice of a 0.8 correlation threshold was somewhat arbitrary, but it represents a high degree of spatial coherence, providing confidence in the robustness of identified patterns. Sensitivity tests conducted for the Mediterranean Sea (Supplementary Figure 4 of Keerthi et al., 2020), varying the threshold between 0.75 and 0.85, showed that our results were only weakly sensitive to changes within this range. This indicates that the findings and interpretations remain consistent across slightly different correlation thresholds.

- ESM future simulations: It could be mentioned why the future simulations analysis was limited from 2084 to 2100, rather than a longer time range.

We used a consistent 16-year time span for all analyses, including historical and future simulations. The satellite SChl observations and historical simulations only share a 16-year common period (1998–2014).

- Use of a single ensemble member: The study currently uses one single ensemble member per ESM. It would be interesting to discuss the implications of this choice, as utilizing the ensemble mean could provide a more accurate representation of model

performance and reduce variability introduced by individual simulations. Similarly, where possible, it would be valuable to discuss the mean across ESMs, as ensemble means often yield more accurate representations than individual models.

We agree with the reviewer that ensemble means often provide more accurate representations than individual models. However, the primary objective of this study is to evaluate the ability of each model to simulate temporal variability in SChl, rather than to identify the best-performing model. This approach is intended to offer insights that modeling groups can use to enhance their models further.

**Technical corrections:**

Ln 81: Umlaut on Müller

Corrected

Ln 84: …more than 'three' times...

Corrected the sentence ' - MPI-ESM1.2-HR has a horizontal resolution twice as high for the atmospheric component (100 km) and more than three times as high for the oceanic component (~40 km) compared to MPI-ESM1.2-LR (200 km and 150 km for the atmospheric and oceanic components, respectively)'.

Ln 88: Keerthi et al. (2022) (comma is not necessary)

Corrected – 'We utilised the datasets outlined in Keerthi et al. (2022) for observed SChl and SST. The SChl data is the Level 3 Mapped 9x9 km resolution 8-day averaged product (release 4.1), covering the period from January 1998 to December 2014'.

Table 1: A border line is missing between HAMOCC6 and 150 km

Corrected

| CMIP6 Simulations | Physical Ocean Model | Ocean BGC Model | Horizontal resolution (Physical & BGC Model) | Model Simulations | References |
|---|---|---|---|---|---|
| IPSL-CM6A-LR | NEMO-OPA | PISCES | 100 km | Historical | Boucher et al. 2018, 2021; Séférian, 2018 |
| IPSL-CM6A-LR-INCA | | | | | |
| CNRM-ESM2-1 | | | | | |
| CESM2 | POP2 | MARBL | 100 km | Historical | Danabasoglu, 2019a, b, c; |
| CESM2-FV2 | | | | | |
| CESM2-WACCM-FV2 | | | | | |
| MPI-ESM1.2-HAM | MPIOM | HAMOCC6 | 150 km | Historical | Neubauer et al., 2019 ; Wieners et al., 2019a, b, c; Jungclaus et al., 2019a, b; Schupfner et al., 2019 |
| MPI-ESM1.2-LR | | | | Historical, SSP5-8.5, piControl | |
| MPI-ESM1.2-HR | | | 40 km | | |
| NorESM2-LM | MICOM | HAMOCC | 100 km | Historical, SSP5-8.5 piControl | Seland et al., 2019a, b, c ; Bentsen et al., 2019a, b |
| NorESM2-MM | | | | Historical, SSP5-8.5 | |

**Table 1.** The CMIP6 Earth system models used in this study; their individual components used to represent ocean and marine biogeochemistry; nominal horizontal resolutions of their ocean and marine biogeochemical models; simulations that were assessed.

Ln 126: Reference the CDO remapping tool remapdis (see reference on: https://code.mpimet.mpg.de/projects/cdo/wiki/Cite)

Thank you for noting it. The citation will be added .Schulzweida, Uwe. (2023). CDO User Guide (2.3.0). Zenodo.https://doi.org/10.5281/zenodo.10020800

Ln 173: 'display' in plural instead of displays

The sentence has been modified – 'CESM2, CNRMESM2-1, and IPSL-CM6A-LR display varying biases relative to satellite SChl across regions'.

Ln 182: Perhaps mention the metric of correlation employed, I assume the Pearson Correlation coefficient?

The sentence has been modified –'The spatial correlation (pearson correlation) between CMIP6 models and observations remains below 0.6 (Fig. 2a), with MPI models showing particularly low correlations, below 0.2'.

Fig 2: Add degree symbol at 60°N and 60°S. For clarity, consider adding to the description that the dots represent models, while dashed lines represent observations. Additionally, complement the color scheme by using different symbols for each model to improve accessibility for color-blind readers.

The Figure 2 and caption has been modified.

[Figure]

Figure 2: Evaluation of the mean spatial distribution. Taylor diagram for the annual mean (a) SChl and (b) SST over the period 1998–2014, within the domain 60°N–60°S. The dashed curve represents the standard deviation of the observational data.

Ln 216: There is no reference to Figure 3 in the text. It would improve clarity to reference Figure 3 here.

The sentence will be modified to 'The variability of SChl across different timescales varies significantly among the CMIP6 simulations (Figure 3)'.

Fig 3: The description states '(Left Panel)', however, I do not see a left and right panel nor a reference to a '(Right Panel)'. Consider adding for clarity: Normalized standard deviation 'of globally averaged' Schla…

Sorry for the confusion. This will be corrected. In the initial stage of the manuscript, Figures 3 and 4 are combined as the left and right panels.

Ln 266-267: Consider adding a reference to the figures in the sentence: The standard deviation across different timescales and the relative contribution of these timescales to the total SST variance '(Figure 4)' show distinct patterns compared to SChl '(Figure 3)'.

It will be corrected as suggested by the reviewer. Thank you for noting it.

Ln 273: The term 'ENSO' is used as an abbreviation without prior introduction. Additionally, 'El Niño' is mentioned in line 469. For coherence and clarity, consider introducing the term in full as 'El Niño–Southern Oscillation' upon its first use, then consistently using either 'El Niño' or 'ENSO' throughout the rest of the manuscript.

We will modify line 273 to introduce ENSO as "El Niño–Southern Oscillation" and will change the term "El Niño" in line 469 to ENSO.

Description Fig 4: Is it standard practice to reference a previous figure or would it be helpful to include the full description again?

Figure 4 caption will be modified as shown below.

Figure 4: Variability across timescales for SST: Similar to Figure 3, but for SST. (Left Panel) (a) Normalised standard deviation of SST from observations and CMIP6 historical simulations. Standard deviation at each grid point is normalised by the mean over each grid. (b) Percentage of SST variance explained by each component (sub-seasonal, seasonal and multiannual) for observations and CMIP6 historical simulations. Shading represents the different model groups described in Section 3.2, with green for Group 1, pink for Group 2, and blue for Group 3. Note that NorESM2-LM and NorESM2-MM are excluded, as daily resolution SST data for these models is not available on the CMIP6 data portal.

Fig 6 ln 353: 800 km in lowercase

Corrected

Ln 473-474: Consider adding the following: The simulated change of the sub-seasonal variability of SChl 'in response to X',…

The following sentence will be modified as 'Specifically, the simulated impact of climate change on the sub-seasonal variability of SChl has, to the best of our knowledge, not been previously assessed in CMIP-type models.